# Optimization and Validation of the GC/FID Method for the Quantification of Fatty Acids in Bee Products

**Laurynas Jarukas** [1], **Greta Kuraite** [1], **Juste Baranauskaite** [2], **Mindaugas Marksa** [1], **Ivan Bezruk** [3] and **Liudas Ivanauskas** [1,*]

1   Department of Analytical and Toxicological Chemistry, Lithuanian University of Health Sciences,
    A. Mickeviciaus str. 9, LT-44307 Kaunas, Lithuania; laurynas.jarukas@lsmuni.lt (L.J.);
    greta.kuraite@gmail.com (G.K.); mindaugas.marksa@lsmuni.lt (M.M.)
2   Department of Pharmaceutical Technology, Faculty of Pharmacy, Yeditepe University Atasehir, Inonu Mah.,
    Kayısdagı Cad., 34755 Istanbul, Turkey; baranauskaite.juste@gmail.com
3   Department of Pharmaceutical Chemistry, National University of Pharmacy, Valentynivska, str. 4,
    461168 Kharkiv, Ukraine; vania.bezruk@gmail.com
*   Correspondence: liudas.ivanauskas@lsmuni.lt; Tel.: +370-673-39-488

**Featured Application: Authors are encouraged to provide a concise description of the specific application or a potential application of the work. This section is not mandatory.**

**Abstract:** To the best of our knowledge, so far, no study has been conducted about the comparison of the total fatty acid concentration in the four bee products (honey, bee pollen, bee bread, and propolis) collected from Lithuania. Therefore, we aimed to optimize the derivatization parameters and to investigate a simple and sensitive gas chromatography-flame ionization detection (GC-FID) method to determine fatty acids. The optimal derivatization parameters were used to analyze fatty acids in the bee products. Regarding sample preparation, three derivatization parameters were compared (temperature and extraction time with BF3/MeOH reagent) in order to obtain a high amount of the total fatty acids of interest from the fatty acid methyl ester (FAME) standard. The results showed that the highest total yield of fatty acids was conducted by using the conventional heating process at 70 °C for 90 min. Under optimal conditions, there was obtained excellent linearity for fatty acids with determination coefficients of $r^2 > 0.9998$. The LODs and LOQs ranged from 0.21 to 0.54 µg/mL and 0.63 to 1.63 µg/mL, respectively. This method has been successfully applied to the qualitative analysis of fatty acids in bee products. The above findings might provide a scientific basis for evaluating the nutritional values of bee products.

**Keywords:** FAME; bee bread; bee pollen; honey; propolis

## 1. Introduction

Fatty acids are crucial in the growth and development of the human body. They are structural components of lipids, which compose membranes of the cells and play an important role in processes like gene expression and cellular communication [1]. Fatty acids are classified into saturated and unsaturated carboxylic acids; their carbon chain can vary from 2 to 36 carbon atoms [2].

Omega-3 and omega-6 are polyunsaturated fatty acids (PUFAs) and they are essential for humans; they cannot be synthesized in our gastrointestinal tract, they must be consumed with food. Anthropological studies indicate that human beings evolved on a diet containing about 1:1 to 1:4 ratio of omega-3 and omega-6 fatty acids, yet over the past century there has been a significant increase of omega-6 intake due to increased consumption of vegetable oils (soybean, sunflower seeds, cottonseed, corn, and safflower seeds), resulting in omega-3 and omega-6 ratio of 1:15–20. According to multiple studies, excessive intake of omega-6 promotes blood clotting and platelet aggregation. Meanwhile, omega-3 fatty acids can

reduce inflammation, which is usually at the base of many chronic diseases; moreover, it can reduce inflammatory effects associated with excessive intake of omega-6. Omega-3 has hypolipidemic, antithrombotic, vasodilatory, and antiarrhythmic effects. The most important for human health omega-3 acids are α-linolenic acid (ALA), eicosapentaenoic acid (EPA), and docosahexaenoic acid (DHA) [3–5].

Deficiency of omega-3 consequently is a deficiency of DHA, which is a long-chain form of omega-3. Deficiency of DHA combined with incorrect omega-3 and omega-6 ratio leads to cognitive disorders and often to mental diseases [6]. In order to obtain the right amount of omega-3, daily diet can be replenished with food supplements or with functional food.

There are studies proving that fatty acids present in pollen enhance the cognitive function of bees and bumblebees, but limited information about fatty acids in bee products [7,8]. There are also studies, proving the presence of polyunsaturated fatty acids (PUFAs) in pollen [3,6]. Our aim is to determine the composition of fatty acids in bee products such as bee pollen, bee bread, honey, and propolis. Such study gives possibility to estimate whether pollen, composing bee products, collected from non-specific plants and in different areas vary significantly in composition of fatty acids.

The selecting the analytical methodology to determine the fatty acids strongly depends on its nature. There are several methods and techniques described for the analysis of fatty acids in bee products. The extraction technique mainly depends on the analytical method that is used during the research. Usually lipids are extracted by different organic solvents and mixtures of it (petroleum ether, methanol, and chloroform), Soxhlet, supercritical fluid extraction, and derivatization. Moreover, while using the supercritical fluid extraction, the fatty acids determination is performed by liquid chromatography. The total fatty acids content has mostly been determined by Soxhtlet extraction followed by gravimetry, while quantification has always been performed spectrophotometrically. Gas chromatography (GC) is the technique of choice when establishing fatty acids following extraction with organic solvents and conversion to the corresponding derivatives [9]. Many different methylation methods are described in literature, and four of them are commonly used: acid or base-catalyzed methylation, borontrifluoride methylation, methylation with diazomethane, and silylation [10]. GC/MS is the most widely used technique for determining the fatty acids profiles, due to its sensitivity and efficacy [10].

GC analysis was performed in order to evaluate and compare the composition of fatty acids in different bee products. Preparation of samples for analysis by GC involves lipid extraction and methylation procedures. During these procedures, derivatization of fatty acids is performed, which transforms fatty acids into fatty acid methyl esters. Fatty acid methyl esters are easier to analyze: fatty acids are highly polar and tend to form hydrogen bonds, which leads to absorption issues.

## 2. Materials and Methods

### 2.1. Materials

Bee products like pollen, bee bread, propolis, and honey were obtained from a beekeepers farm "Viliaus Rinkūno ūkis", Lithuania. The bee products like pollen, bee bread, propolis, and honey were collected in Katiliu countryside, Sakiu district, Lithuania (54° 52′ 30″ N, 23° 8′ 20.4″ E 54.875°, 23.139°). The samples were obtained in 2018 after the harvesting season (July/September). Voucher specimens (No. 1359352) have been deposited at the Herbarium of the Department of Analytical and Toxicological Chemistry, Lithuanian University of Health Sciences.

### 2.2. Solvents and Reagents

The water used for sample preparation was produced with a Super Purity Water System (Millipore, Burlington, MA, USA). Standards for GC analysis: Supelco 37 Component fatty acid methyl ester (FAME) Mix (North Harrison Road, Bellefonte, PA, USA) purity of fame mix components not less than 98.7%. Methanol (99.9%), chloroform (99.9%), hexane

(>95%), toluene (99.7%) boron trifluoride-methanol solution 10% (1.3 M) were purchased from Sigma–Aldrich Co., UK.

### 2.3. Extraction of Fats from Bee Products

The extraction of fats from all bee products (0.5 g) was accomplished by using 10 mL of chloroform/methanol mixture (1:1) and 1000 μL of water, then the prepared samples were sealed and left to stand overnight in a dark place at 20 °C ± 2 °C. The 1 mL of chloroform layer was transferred into another tube and the solvent was removed by evaporation process. The fatty acid esters were hydrolyzed and methylated simultaneously with a mixture of 100 μL of toluene and 0.5 mL of boron trifluoride/methanol (BF3/MeOH) for 90 min at 70 °C by using glycerol bath. After cooling, 800 μL of distilled water and 800 μL of hexane were added. After shaking and settling, the hexane layer (upper layer) containing fatty acid methylated esters (FAME) was transferred to gas chromatography (GC) vials and carried in −4 °C degree until analysis.

### 2.4. Quantitative and Qualitative Analyses

GC/FID was performed according to the certified methodology, which was specified with standard solution Supelco 37 Component FAME Mix. Gas chromatography with flame ionization detector (FID) analysis was performed on the Shimadzu GC-2010 Plus. Analytical conditions: volume injected, 1 μL; carrier gas helium, 1.26 mL/min; injector temperature, 230 °C; flame ionization detector temperature 250 °C; split ratio, 1:20; and oven temperature ranged from 100 to 240 °C with a stepwise temperature program within the total run time of 71.67 min. For analysis we used a 100 m Restek RT2560″ column; diameter: 0.25 μm; thickness: 0.20 μm.

### 2.5. GC/FID Method Validation

Validation of the GC/FID method was performed according to the international guidelines on analytical techniques for quality control of pharmaceuticals (ICH guidelines) [11]. Method validation was performed to assess linearity and ranges of fatty acids calibration curves. Regarding linearity, a standard of fatty acids solution mix was prepared as follows. An accurate volume of standard solution was placed into a volumetric flask (399.80 μg/mL). The precision of the method was evaluated by calculating repeatability (r). The precision of extraction technique was validated by repeating extraction procedure with standard mix solutions six times. An aliquot of each extract was then injected and quantified. The precision of a chromatographic system was tested by checking the %RSD of retention times and peak areas. Six injections were performed each day for three consecutive days.

### 2.6. Statistical Analysis

The results were analyzed by one-way analysis of variance (ANOVA) followed by Tukey's multiple comparison test with software package Prism v. 5.04 (Graph Pad Software Inc., La Jolla, CA, USA). We estimated the average of measurement (AVG), sample standard deviation (Sx), a standard deviation of mean (SD), the coefficient of variation (CV), and the statistical significance of results (*p*). The value $p < 0.05$ was taken as statistically significant. The correlation and regression analysis were performed for evaluation of impact of clover pollen on the content of fatty acids in the samples.

## 3. Results and Discussion

### 3.1. Methylation Parameters

Before GC analysis, the fatty acid components of lipids are converted to the simplest convenient volatile derivative, usually methyl esters [12]. The Lewis acid, boron trifluoride, in the form of its coordination complex with methanol, is a powerful acidic catalyst for the transesterification of fatty acids. The primary advantage of acid catalysis is the general applicability, with both bound and free fatty acids (FFA) being converted concurrently to FAME. Among the various acid-catalyzed reagents (such as methanol-hydrochloric

acid, sulfuric acid in methanol, and acetyl chloride in methanol), boron trifluoride in methanol has a wide application as a good reagent to convert both the acyl-glycerols and the FFA into methyl esters [12]. Extraction temperature and duration are the most important factors contributing to the yield of FAMEs. To select a suitable temperature for derivatization method, we performed the FAME reaction at different temperatures (from 60 to 100 °C) with 30 min as fixed reaction time. The results of using five different temperatures (Figure 1) showed that the optimum temperature was 70 °C, with a significant difference in yield of all FAMEs ($P < 0.05$) in comparison with other temperatures. It was revealed that increasing temperature significantly decreased the concentration of fatty acids in the samples. This might be explained by fatty acids degradation during the methylation process. The optimum derivatization temperature by using conventional heating found from this study (70 °C) was in good agreement with other studies [13]. Although temperatures higher than 100 °C may provide equal or higher yields of FAMEs, the operation of derivatization above this temperature was not possible due to sample damage. Moreover, this can be explained because we used methanol, and high temperature may cause loss of solvent (the boiling point of methanol is 60 °C) as the vials were not completely gastight and transfer of double bond may have occurred. The temperature at 70 °C was, therefore, selected as an optimum point and used for further investigation.

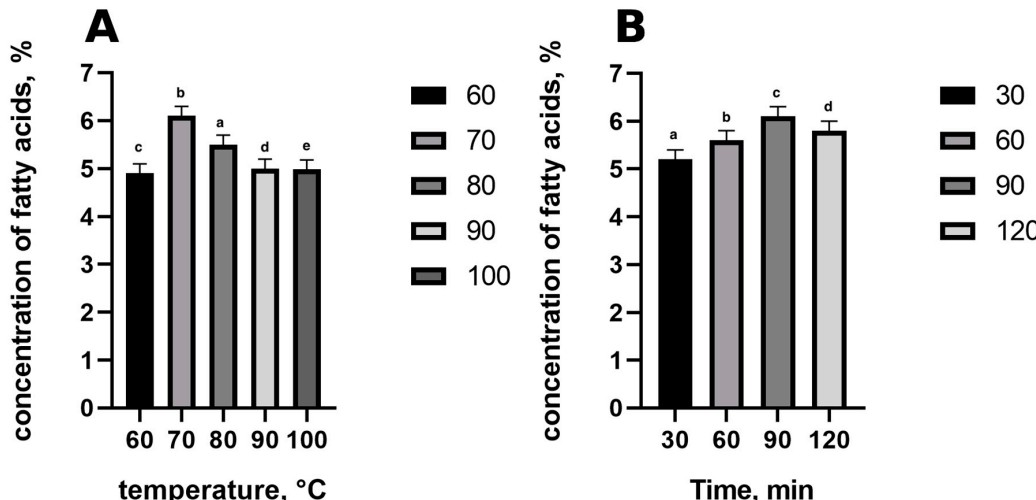

**Figure 1.** Effect of temperature (**A**) on derivatization reaction. Effect of time (**B**) on derivatization reaction; all reactions were performed at 70 °C. Results are expressed as means ± standard error, (*n* = 6). Values within columns followed by the same lowercase letter (a–d) differed statistically at *p* < 0.05 (Tukey's test).

To identify the optimal reaction time, we performed conventional heating derivatization between 30 to 120 min (Figure 1) by using fixed temperature at 70 °C, the results showed significant difference between the tested conditions (*p* < 0.05). Moreover, the reaction time of 90 min was sufficient to complete the conversion of all fatty acids. The optimum time for FAME preparation may also depend on the type and size of sample. Samples with complex matrices may require a longer time for a complete lipid extraction [12].

In conclusion, the preparation procedure of FAME for GC analysis without prior lipid extraction via the one-step method has a number of advantages as it is simple, time-saving, reduces the use of solvents, and minimizes the potentials for sample loss and contamination. By combination of the extraction–methylation processes, FAME preparation can be completed within 90 min and ready for GC analysis.

*3.2. Method Validation*

Qualitative analysis was performed using Supelco 37 Component FAME Mix" standard, the components identified by retention times and profile of chromatogram in the certificate of standards mix (Table 1).

**Table 1.** Linearity and homoscedasticity test for calibration plots, results of the system suitability study, and detection and quantitation limit values for fatty acids methyl esters.

| Nr. | FAME | RT (Mean) | $R^2$ | Range (µg/mL) | Detection (LOD), µg/mL | Quantitative (LOQ), µg/mL |
|-----|------|-----------|-------|---------------|------------------------|---------------------------|
| 1. | Methyl butyrate | 10.794 | 0.999 | 6.24–399.6 | 0.51 | 1.54 |
| 2 | Meethyl hexanoate | 12.866 | 0.999 | 6.24–399.6 | 0.44 | 1.33 |
| 3 | Methyl octanoate | 16.514 | 0,999 | 6.24–399.6 | 0.42 | 1.28 |
| 4 | Methyl decanoate | 21.548 | 0.999 | 6.24–399.6 | 0.39 | 1.18 |
| 5 | Methyl laurate | 27.024 | 0.999 | 6.24–399.6 | 0.33 | 1.01 |
| 6 | Methyl tridecanoate | 29.679 | 0.999 | 3.08–197.5 | 0.32 | 0.98 |
| 7 | Methyl myristate | 32.263 | 0.999 | 6.24–399.6 | 0.29 | 0.89 |
| 8 | Myristoleic acid methyl ester | 34.256 | 0.999 | 3.12–199.8 | 0.31 | 0.93 |
| 9 | Methyl pentadecanoate | 34.706 | 0.999 | 3.12–199.8 | 0.28 | 0.84 |
| 10 | Cis-10-pentadecanoic acid methyl ester | 36.649 | 0.999 | 3.09–198.0 | 0.28 | 0.86 |
| 11 | Methyl palmitate | 37.09 | 0.999 | 9.36–599.4 | 0.25 | 0.76 |
| 12 | Methyl palmitoleate | 38.629 | 0.999 | 3.12–199.8 | 0.27 | 0.81 |
| 13 | Methyl heptadecanoate | 39.292 | 0.999 | 3.11–199.2 | 0.31 | 0.94 |
| 14 | Cis-10-heptadecanoic acid methyl ester | 40.815 | 0.998 | 3.12–199.8 | 0.24 | 0.72 |
| 15 | Methyl stearate | 41.474 | 0.999 | 6.24–399.6 | 0.22 | 0.68 |
| 16 | Trans-9-elaidic acid methyl ester | 42.343 | 0.998 | 3.12–199.8 | 0.21 | 0.64 |
| 17 | Cis-9-oleic acid methyl ester | 42.757 | 0.999 | 6.24–399.6 | 0.24 | 0.71 |
| 18 | Methyl linoleate | 44.667 | 0.999 | 3.09–197.9 | 0.25 | 0.76 |
| 20 | Methyl arachidate | 45.518 | 0.998 | 6.24–399.6 | 0.21 | 0.63 |
| 21 | Gamma-linolenic acid methyl ester | 46.089 | 0.999 | 3.12–199.8 | 0.25 | 0.77 |
| 22 | Methyl eicosanoate | 46.672 | 0.999 | 3.11–199.2 | 0.21 | 0.65 |
| 23 | Methyl linolenate | 46.854 | 0.999 | 3.12–199.8 | 0.27 | 0.81 |
| 24 | Cis-11,14-eicosadienoic acid methyl ester | 48.475 | 0.999 | 3.12–199.8 | 0.23 | 0.69 |
| 26 | Methyl behenate | 49.268 | 0.998 | 6.23–398.8 | 0.26 | 0.79 |
| 30 | Cis-11,14,17-eicotrienoic acid methyl ester | 50.507 | 0.999 | 3.03–193.6 | 0.27 | 0.82 |
| 33 | Cis-13,16-docosadienoic acid methyl ester | 52.014 | 0.999 | 3.12–199.8 | 0.28 | 0.84 |
| 34 | Methyl lignocerate | 52.786 | 0.999 | 6.24–399.6 | 0.26 | 0.79 |
| 35 | Methyl cis-5,8,11,14,17-eicosapentaenoate | 52.949 | 0.999 | 3.12–199.8 | 0.33 | 1.01 |
| 36 | Methyl nervonate | 53.843 | 0.999 | 3.12–199.8 | 0.29 | 0.87 |
| 37 | Cis-4,7,10,13,16,19-docosahexaenoic acid methyl ester | 57.949 | 0.999 | 3.12–199.8 | 0.54 | 1.63 |

Mean of six replications. RT: retention time. R: correlation coefficient. LOD: limit of detection, LOQ: limit of quantifiction.

System suitability was evaluated by replicate ($n = 6$) injection of the same standard solution containing FAMEs. The method validation for fatty acids was performed by analyzing the repeatability and checking the precision of %RSD of retention times and peak areas. After all, the repeatability for standards calculating %RSD for retention times was not greater than 0.5% and for peak areas not greater than 1.0%, calculating the same conditions for the precision of retention times was not bigger than 0.5%, for peak areas are not greater than 1.5%.

Quantitative analysis of fatty acids was performed by evaluating the values of analyzed fatty acids and using calibration curves. The concentration on the standards in the mixture of fatty acids was between 3.09 and 599.40 µg/mL; the calibration curves consisted of seven concentrations. The correlation coefficient was no less than $r^2 = 0.999$, thus confirming the linearity of the method (Table 1).

The LOD was determined as three times the signal-to-noise ratio, while the LOQ was ten times the signal-to-noise ratio. The LOD and the LOQ values for different FAMEs are reported in Table 1. Accuracy and precision of the method were determined by replicate analysis ($n = 6$) of FAME reagent, which is reported in Table 1. The LOD values varied between 0.21 µg/mL to 0.54 µg/mL, while the LOQ between 0.64 µg/mL to 1.63 µg/mL (Table 1), which indicates that the method is sensitive. According to the described data above, it can be concluded that this method is a reliable tool for the identification and quantification of fatty acids in bee products, conforming to the ICH guidelines [11].

### 3.3. Determination of Total Fatty Acids in Bee Products

Fatty acids play an essential role in human diet and health. A high amount of lipids may provide abundant fatty acids. The fatty acid composition of bee products mostly depends on the botanical and geographical origin, as well as on the used methodology for isolation and extraction of fatty acids [13,14]. The method developed in this study was employed for the determination of saturated/unsaturated fatty acids, and the ratio of total unsaturated/saturated fatty acid (TUS/TS) in four different bee products (honey, bee pollen, bee bread, and propolis). All investigated fatty acids were identified by their retention times. The typical chromatogram of the bee products was shown in Figure 2, and fatty acids contents were summarized in Figure 3.

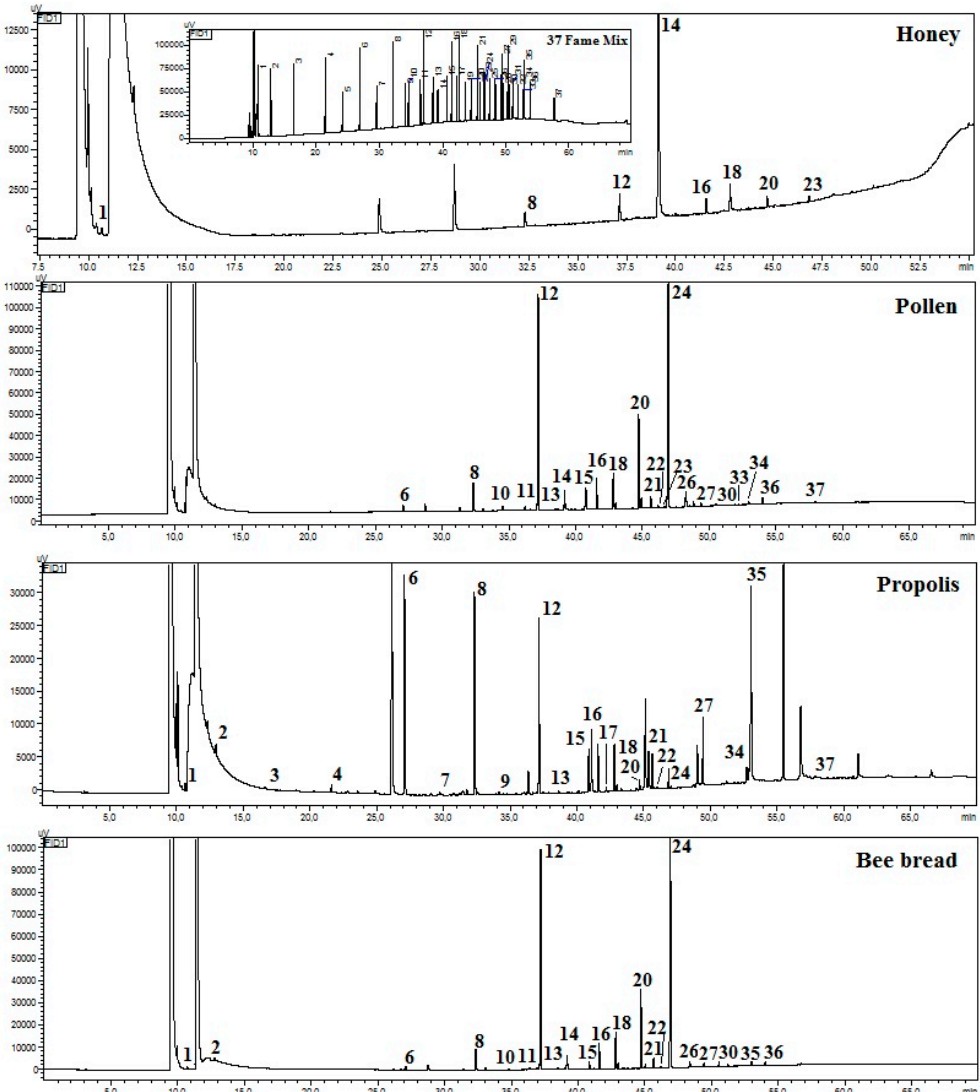

**Figure 2.** Gas chromatography-flame ionization detection (GC-FID) chromatograms of fatty acid methyl ester (FAME) standard and fatty acids derivatives extracted from honey, bee pollen, bee bread, and propolis. The obtained fatty acids derivatives are marked as 1—Methyl butyrate; 2—Meethyl hexanoate; 3—Methyl octanoate; 4—Methyl decanoate; 5—Methyl laurate; 6—Methyl tridecanoate; 7—Methyl myristate; 8—Myristoleic acid methyl ester; 9—Methyl pentadecanoate; 10—Cis-10-pentadecanoic acid methyl ester; 11-Methyl palmitate; 12—Methyl palmitoleate; 13—Methyl heptadecanoate; 14—Cis-10-heptadecanoic acid methyl ester 15—Methyl stearate; 16—Trans-9-elaidic acid methyl ester; 17—Cis-9-oleic acid methyl ester; 18—Methyl linoleate; 20—Methyl arachidate; 21—Gamma-linolenic acid methyl ester; 22—Methyl eicosanoate; 23—Methyl linolenate; 24—Cis-11,14-eicosadienoic acid methyl ester; 26—Methyl behenate; 30—Cis-11,14,17-eicotrienoic acid methyl ester;33—Cis-13,16-docosadienoic acid methyl ester; 34—Methyl lignocerate; 35—Methyl cis-5,8,11,14,17-eicosapentaenoate; 36—Methyl nervonate; 37—Cis-4,7,10,13,16,19-docosahexaenoic acid methyl ester.

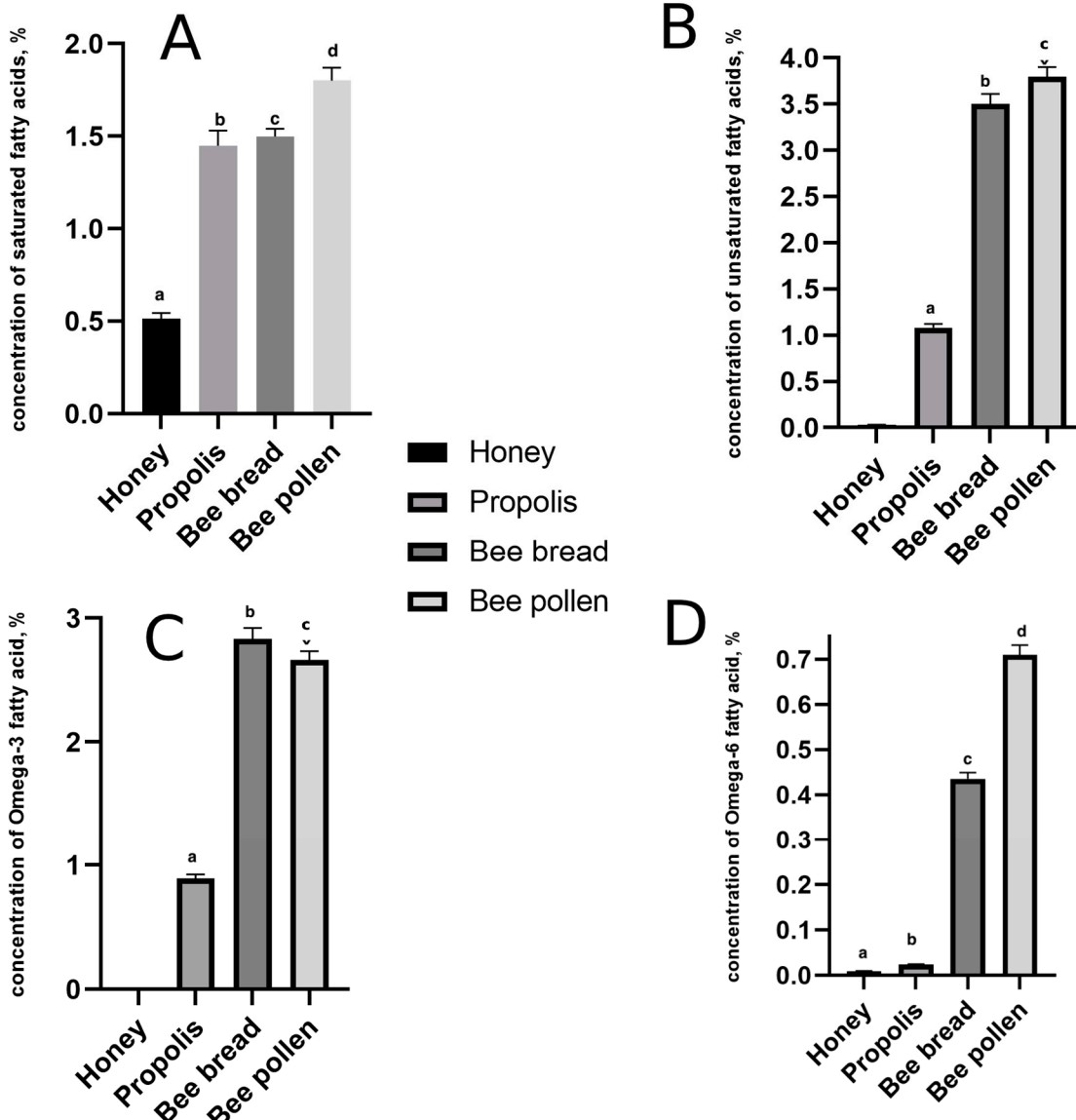

**Figure 3.** The quantitative amount (expressed in percentage) of fatty acid and composition ((**A**) Total Saturated Fatty Acid, (**B**) Total Unsaturated Fatty Acid, (**C**) Total Omega-3, and (**D**) Omega-6 Fatty Acid) in propolis, bee bread, bee pollen, and honey, (*n* = 6). Different letters in each column denote statistical difference at $p \leq 0.05$.

Qualitative analysis of fatty acids was performed by evaluating the chromatograms of all products and recording the retention times of fatty acids compared to the standard chromatogram, thus identifying the number of different fatty acids in the test products. The results of the qualitative analysis of fatty acids are presented in Table 2. The results of the qualitative analysis of fatty acids are presented in Table 2. It was found that the total number of fatty acids in bee pollen was 22, in bee bread was 21, in propolis was 22, and in honey was eight (Table 2). The highest number of TSFA was found in propolis (12 RR) and lowest – in honey (6 RR). The highest number of TUSFA (14 RR) was found in pollen, and the lowest (2 RR) in honey (Table 2). One trans-fatty acid was found in propolis. After comparing results of the research with the data obtained by other scientists, it was noticed that Lithuanian scientists have detected 42 fatty acids in bee pollen and 39 RR in bee bread. However, the marked differences of results are due to a fact that the study conducted by these researchers used the FAME standard, which contains more than 50 different fatty acid methyl esters.

**Table 2.** Fatty acid content, ratios of total unsaturated (TUS) to saturated (TS) fatty acids, and omega-6 to omega-3 fatty acids in propolis, beebread, bee pollen, and honey.

| Bee Product | Total Number of Saturated Fatty Acids | Total Number of Unsaturated Fatty Acids | Omega-6/Omega-3 | TUS/TS |
|---|---|---|---|---|
| Propolis | 10 | 12 | 36.6:1 | 0.74:1 |
| Bee bread | 10 | 11 | 6.52:1 | 2.29:1 |
| Bee pollen | 12 | 9 | 3.79:1 | 2.07:1 |
| Honey | 6 | 2 | 0 | 0.06:1 |

The amount of total saturated fatty acids (TSFA) varied between 0.51 to 1.82% in bee products (Figure 3). The highest yield of TSFA was from bee pollen, whereas the lowest yield was from honey. The obtained yield of TSFA in bee pollen was in agreement with the literature [15]. Moreover, it could be considerable differences in fat composition of bee pollen, depending on the botanical origin and according to the literature; the TSFA yield could vary between 1 to 13 g/100 g dry weight [16]. Moreover, no statistical difference was revealed among bee bread and propolis.

Moreover, in this part of the study, the total percentage of unsaturated (omega-3, -6, -9, -7, and -5) fatty acids in the products, excluding unsaturated fatty acids with trans configurations, was evaluated and presented in Figure 3. The amount of total unsaturated fatty acid (TUSFA) varied between 0.03% and 3.76% in bee products. The highest amounts of TUSFA were found in bee pollen (3.76 ± 0.11%) and bee bread (3.53 ± 0.12%), and the lowest (0.03 ± 0.001%) fatty acid content was determined in honey. The total percentage of fatty acids belonging to the omega-3 fatty acid group in the test products was evaluated. The standard used in the study included four omega-3 fatty acids (in the form of methyl esters): α-linolenic, cis-11, 14, 17-eicosatriene, cis-5,8,11,14,17-eicosapentaene (EPR), and cis-4,7,10,13,16,19-docosahexaenoic (DHR) fatty acids. It was observed that polyunsaturated omega-3 fatty acids were not detected in honey. In other studied products, the content of omega-3 fatty acids varied from 0.89 to 2.66%. The highest amounts of omega-3 fatty acids were found in bee bread (2.83 ± 0.09%) and bee pollen (2.66 ± 0.08%), and the lowest – in propolis (0.89 ± 0.03%). Moreover, the total percentage of fatty acids belonging to the omega-6 fatty acid group in the bee products was evaluated too. The standard used in this study included six omega-6 fatty acids (in the form of methyl esters): linoleic, gamma-linolenic, cis-11,14-eicosadienoic, cis-8,11,14-eicosatriene, cis-5,8,11,14-eicatosetraenoic, and cis-13,16-docosadiene fatty acids. It was found that the total yield of polyunsaturated omega-6 fatty acids in bee products varied between 0.01% and 0.70% in the bee products. The highest amount of omega-6 fatty acids was found in bee pollen (0.70 ± 0.02%) and the lowest (0.01–0.02%) in propolis and honey. Furthermore, during the studies, it was found that the ratio of omega-3/omega-6 fatty acids in the investigated products varied from 3.79:1 to 36.6:1. The highest ratio was found in propolis (36.6:1), and the lowest in bee pollen (3.79:1). The ratio of omega-3/omega-6 fatty acids in honey was not determined, possibly due to a negligible amount of omega-3 fatty acids. In bee pollen, the omega-3/omega-6 ratio was 3.79:1, and in bee bread was 6.52:1. The obtained results are in agreement with other authors [17]. Čeksterytė et al. found a lower omega-3/omega-6 ratio in bee pollen—3.35:1, but slightly higher in bee bread—8.42:1 [17]. Unfortunately, none of the tested products had a recommended 1:1 to 1:4 ratio of omega-3/omega-6 ratio dose for humans use. According to the literature, nowadays, eating habits of people have changed and the predominant ratio of omega-3/omega-6 reached 1:20 in daily diets [18]; therefore, the greater amount of omega-3 fatty acids obtained in bee products could be helpful in adjusting this ratio for higher intakes of omega-3 fatty acids.

The ratios of TUS/TS were more than 1.0, with exception of honey, which characteristically lacked TUSFA (0.03 ± 0.0002%) while having the highest concentration of TSFA (0.51 ± 0.002%). The highest ratios of TUS/TS were found in bee bread (2.29 ± 0.02%) and in pollen (2.19 ± 0.03%). The results could be explained by the similar chemical composition of bee bread and pollen [19]. The results have also consisted of notion that

bees collect pollen with a high level of unsaturated fatty acids and the bee bread mainly includes pollen, honey and secretions of bee's salivary glands [20]. Moreover, Lithuanian scientists have found that bee pollen and bread have a lower ratio of TUS/TS 1.58 and 1.76, respectively [19]. There were considerable variations in the TUS/TS ratio, which might have contributed to the different botanical origins or the processing and storage conditions. Furthermore, following Patient Safety Organization (PSO) recommendations saturated fatty acids should be replaced with unsaturated fatty acids [18], due to this reason, the ratio of TUS/TS obtained in bee bread and bee pollen is beneficial, it is recommended to use these products as nutritional food supplements to increase unsaturated fatty acids amount in a diet for health benefits. Moreover, the total percentage of fatty acids belonging to the group of unsaturated trans-fatty acids in the bee products was estimated. Standard used in the study included two trans-fatty acids (in the form of methyl esters): trans-9-elaidic and linoleic fatty acids. During the study, it was observed that trans-fatty acids were detected only in propolis $0.008 \pm 0.00004\%$, but at a very low level that does not pose a risk to human health.

## 4. Conclusions

A selective and sensitive method for determination of fatty acids has been developed and validated in the present work. The method consists of sample preparation, derivatization, and chromatographic analysis. All steps were extensively studied and optimized for the derivatization procedure. In this study, we compared the total fatty acid concentration (saturated, unsaturated, omega-3, omega-6, the ratio of saturated and unsaturated, omega-3/omega-6 fatty acids, and trans-fatty-acids) in four bee products (honey, bee pollen, bee bread, and propolis) collected from Lithuania. The optimal conditions allowed us to reach the highest derivatization efficiency of fatty acids in only 90 min by using conventional heating process. The developed method has been successfully applied to quantification of fatty acids in bee products. This research also shows the interesting agricultural potential of bee products, in relation to the preparation of certified extracts with a high content of fatty acids to be used in the pharmaceutical and nutraceutical areas. In conclusion, the above findings might provide a scientific basis for evaluating the nutritional values of bee products and contribute to a database of food composition.

**Author Contributions:** L.J., G.K., J.B., and M.M. contributed to investigation, data analysis, and original draft preparation. M.M. and J.B. contributed to methodology, data analysis, visualization, and review and editing. L.I. and I.B. contributed to conceptualization, resources, original draft preparation, review and editing, project administration, and supervision. All authors have read and agreed to the published version of the manuscript.

**Funding:** This research received no external funding.

**Institutional Review Board Statement:** Not applicable.

**Informed Consent Statement:** Not applicable.

**Acknowledgments:** The authors are thankful for the financial support provided by Science Foundation of Lithuanian University of Health Sciences.

**Conflicts of Interest:** The authors declare no conflict of interest.

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
