# Peer review of "Optimization and Validation of the GC/FID Method for the Quantification of Fatty Acids in Bee Products"

_applsci, doi:10.3390/app11010083_

Round 1

Reviewer 1 Report

Paper Title: Chemical Composition of Fatty Acids in Bee Products

Review comments

The paper describes a methodology applied to bee products to determine fatty acids.

In my opinion, the paper is suitable to be published bus some comments should be considered and corrected.

Main comments:

  • Title: the title is too general. Authors present a method and study only one sample of each bee product. Composition of bee products, among other factors, depends on the botanical and geographical origin, so that data of only one sample of each product is inadequate to use this title.

I suggest something similar to:

Proposal of a methodology to compare fatty acids composition of bee products

Comparison of fatty acids composition of bee products obtained in Lithuania

  • Table 2 and Table 3 could be merged into a new table. Also, information about the percentage of each type of fatty acid (the information that is in the text) should be added in a new column.

Regarding this information (percentage of the different types of fatty acid), it is not clear what it is the meaning of the percentage. Does it correspond to the full composition of the product? Or is it only a relative percentage of what?

  • In figure 1 and 2, I think it is better to use the same style as figure 4: “Different letters in each column denote statistical difference at p = 0.05.” Anyway, the results or the comments should be checked due to for figure 2 authors comment in lines 146-148,

“To identify the optimal reaction time, we performed conventional heating derivatization between 30 to 120 min (Figure 2) by using fixed temperature at 70 °C, the results showed no significant difference between the tested conditions (p< 0.05)”

In the legend of figure 2, (lines 153-156)

Figure 2. Effect of time on derivatization reaction. All reactions were performed at 70 °C. Results are expressed as means ± standard error, (n=6). Asterisk indicates that the yield of total fatty acids derivatives 90 min and were significantly higher (p < 0.05) than 30, 60 and 120 min. Values within columns followed by the same lowercase letter (a-d) differed statistically at p< 0.05 (Tukey’s test).

The asterisk is missing.

  • In conclusions (lines 311-312).

The conducted research does not allow to extract this affirmation.

Minor considerations

Figure 4. The lowercase of figures B and C for bee pollen is not clear. Is it “e” or “c”

Data in table 3 do not match exactly with data in the text. For example, for propolis 36.6 in the table whereas in the text figures 36.35 and the similar for pollen.

I suggest using only two decimals for all the values in percentage.

Line 57-58, the meaning of this sentence it is not clear. I suggest to delete it. Authors only study one sample of each product.

Line 285-286, the meaning of this sentence is unclear according to information in the text.

The following abbreviations should be extended the first time they were used:

FAME, (line 23)

PUFAS

Voucher specimens (No. 1359352)

Certified methodology

Experimental bee products

PSO

There are some mistakes in references

I hope my comments help you to improve the paper.

Author Response

The authors would like to thank Reviewers for their comments and suggestions to improve the quality of the manuscript. Please find below our response to the remarks, the corresponding corrections of the manuscript have been made. Moreover, the stylistic corrections and additions as well as some new text were added on the basis of recommendations of reviewers.

Comments and suggestions to authors

Title: the title is too general. Authors present a method and study only one sample of each bee product. Composition of bee products, among other factors, depends on the botanical and geographical origin, so that data of only one sample of each product is inadequate to use this title.

I suggest something similar to:

Proposal of a methodology to compare fatty acids composition of bee products

Comparison of fatty acids composition of bee products obtained in Lithuania

Response: the title was changed to Comparison of fatty acids composition of bee products obtained in Lithuania.

·       Table 2 and Table 3 could be merged into a new table. Also, information about the percentage of each type of fatty acid (the information that is in the text) should be added in a new column.

Response: Table 2 and Table 3 was merged into a new table. The mistake was corrected information about the percentage of each type of fatty acid (the information that is in the text) are shown in the figure 4.

·       In figure 1 and 2, I think it is better to use the same style as figure 4: “Different letters in each column denote statistical difference at p = 0.05.” Anyway, the results or the comments should be checked due to for figure 2 authors comment in lines 146-148,

“To identify the optimal reaction time, we performed conventional heating derivatization between 30 to 120 min (Figure 2) by using fixed temperature at 70 °C, the results showed no significant difference between the tested conditions (p< 0.05)”

·        

Response: for figure 1 and figure 2 was used the same style as figure 4. The mistake has been corrected.

Regarding this information (percentage of the different types of fatty acid), it is not clear what it is the meaning of the percentage. Does it correspond to the full composition of the product? Or is it only a relative percentage of what?

Response: the quantitative amount of fatty acids is expressed in percentage.

In the legend of figure 2, (lines 153-156)

Figure 2. Effect of time on derivatization reaction. All reactions were performed at 70 °C. Results are expressed as means ± standard error, (n=6). Asterisk indicates that the yield of total fatty acids derivatives 90 min and were significantly higher (p < 0.05) than 30, 60 and 120 min. Values within columns followed by the same lowercase letter (a-d) differed statistically at p< 0.05 (Tukey’s test).

The asterisk is missing.

Response:   The mistake has been corrected the asterisk has been removed, because the information has shown in the results and discussion section.

·       In conclusions (lines 311-312).

The conducted research does not allow to extract this affirmation.

Response: the sentence has been changed according to the requirements.

Figure 4. The lowercase of figures B and C for bee pollen is not clear. Is it “e” or “c”

Response: the mistake has been corrected.

I suggest using only two decimals for all the values in percentage.

Response: it was changed according to requirements.

Line 57-58, the meaning of this sentence it is not clear. I suggest to delete it. Authors only study one sample of each product.

Response: the sentence was deleted.

Line 285-286, the meaning of this sentence is unclear according to information in the text.

Response: the sentence has been changed: Moreover, Lithuanian scientists have found that bee pollen and bread, have a lower ratio of TUS/TS 1.58 and 1.76 respectively.

The following abbreviations should be extended the first time they were used:

FAME, (line 23)

PUFAS

Voucher specimens (No. 1359352)

Certified methodology

Experimental bee products

PSO

Response: the abbreviations were extended according to the suggestions.

There are some mistakes in references

Response: the mistake has been corrected.

Reviewer 2 Report

The pharmaceutical and nutraceutical properties of bee products are well known, and there are several studies that relate various chemical compounds to the botanical origin of these products. The general topic of the paper is relevance at this moment. However, in reality the title is not in accordance with the content of the paper, the authors actually optimize and validate a method (GC-FID method) for the quantification of fatty acids in bee products. The objective posed by the authors, optimizing the same method for several bee products is an interesting idea; it facilitates preparation tasks, reagents, derivatization and response time, and consequently it is cheaper. I propose to change the title in this sense: `Optimization and validation of the GC/FID method for the quantification of fatty acids in bee products´.

In my opinion, there are some main problems with the manuscript, that does not favor the real contribution of the study.

- In material and methods, it is necessary to specify the number of samples (honey, propolis, bee pollen and bread pollen) used and information on the botanical origin. This information is key for possible relationships with some identified fatty acids.

- The number of samples is limited (N=6), consequently the results can be unreliable because very little data are analyzed and compared.

- In Line 80: for the extraction of fatty acids, 0.5 g is dissolved in 10 mL chloroform/methanol, is this initial concentration the same for all bee products?

-With the proposed method is it possible to do the quantification simultaneously with honey, pollen or propolis?

- Table 1: I understand that the results in the table are analyzed on 6 samples, but are they with all bee products (honey, pollen, propolis)? are the same LOQ and LOD? Is it possible to specify them for each product? Taking into account that the matrix of honey in terms of fatty acids has nothing to do with the rest of the bee products (honey has practically no lipids), it could be interesting to expose the validation of the method for each bee product.

Other minor comments:

Introduction: A compilation of techniques used in the determination of fatty acids is missing, as well as their validity and efficacy in bee products. It would be interesting to include this information in the introduction, since as the authors comment, there are few studies on this line. Perhaps it is motivated because honey is the most analyzed and characterized product, and has a low concentration in lipids. However, in recent years, pollen is gaining ground and there are several published investigations on the chemical composition of this product. But, the comparison of the 4 bee products that the authors propose is fine.

Figure 3: include what is each number (fatty acid).

Line 299-300: I recommend deleting this sentence because with only 6 samples I consider that it is not adequate. As a first approximation it may be, but some aspects are missing in the manuscript (results and discussion) and that are relevant, such as the botanical origin of the analyzed samples, which can corroborate the high values in some compounds and relate them to the specific pollen of the plant.

Author Response

“Optimization and Validation of the GC/FID Method For the Quantification of Fatty Acids in Bee Products”

The authors would like to thank Reviewers for their comments and suggestions to improve the quality of the manuscript. Please find below our response to the remarks, the corresponding corrections of the manuscript have been made. Moreover, the stylistic corrections and additions as well as some new text were added on the basis of recommendations of reviewers.

Comments and suggestions to authors

The pharmaceutical and nutraceutical properties of bee products are well known, and there are several studies that relate various chemical compounds to the botanical origin of these products. The general topic of the paper is relevance at this moment. However, in reality the title is not in accordance with the content of the paper, the authors actually optimize and validate a method (GC-FID method) for the quantification of fatty acids in bee products. The objective posed by the authors, optimizing the same method for several bee products is an interesting idea; it facilitates preparation tasks, reagents, derivatization and response time, and consequently it is cheaper. I propose to change the title in this sense: `Optimization and validation of the GC/FID method for the quantification of fatty acids in bee products´.

Response: the title has been changed according to the suggestions.

In material and methods, it is necessary to specify the number of samples (honey, propolis, bee pollen and bread pollen) used and information on the botanical origin. This information is key for possible relationships with some identified fatty acids.

Response: The bee products like pollen, bee bread, propolis and honey were collected in area of Lithuania: Sakiu district; Katiliu countryside (54° 52′ 30″ N, 23° 8′ 20.4″ E 54.875°, 23.139°). The samples were obtained in 2018 after the harvesting season (July/September).

The number of samples is limited (N=6), consequently the results can be unreliable because very little data are analyzed and compared.

Response: it was used 6 replicates in each sample.

In Line 80: for the extraction of fatty acids, 0.5 g is dissolved in 10 mL chloroform/methanol, is this initial concentration the same for all bee products?

Response: the initial concentration was the same for all bee products.

With the proposed method is it possible to do the quantification simultaneously with honey, pollen or propolis?

Response: the quantification studies of the fatty acids in the honey, pollen, bread and propolis was done. The fatty acids were separated into the groups: unsaturated, saturated, omega-6 and omega-3 fatty acids, the amount of the related fatty acids was calculated (expressed in percentage) in each sample.

Table 1: I understand that the results in the table are analyzed on 6 samples, but are they with all bee products (honey, pollen, propolis)? are the same LOQ and LOD? Is it possible to specify them for each product? Taking into account that the matrix of honey in terms of fatty acids has nothing to do with the rest of the bee products (honey has practically no lipids), it could be interesting to expose the validation of the method for each bee product.

Response: the validation was made with FAMEs standard so LOQ and LOD is made according to the fatty acids’ included into this standard. The aim was to evaluate the differences between all products. We agreed that honey sample has almost no lipids so we calculated the amounts of fatty acids by using the calibration equation.

Introduction: A compilation of techniques used in the determination of fatty acids is missing, as well as their validity and efficacy in bee products. It would be interesting to include this information in the introduction, since as the authors comment, there are few studies on this line. Perhaps it is motivated because honey is the most analyzed and characterized product, and has a low concentration in lipids. However, in recent years, pollen is gaining ground and there are several published investigations on the chemical composition of this product. But, the comparison of the 4 bee products that the authors propose is fine.

Response: the information about the determination of fatty acids and their validity and efficacy was included in introduction part.

Figure 3: include what is each number (fatty acid).

Response: the explanations of the numbers were included.

Line 299-300: I recommend deleting this sentence because with only 6 samples I consider that it is not adequate. As a first approximation it may be, but some aspects are missing in the manuscript (results and discussion) and that are relevant, such as the botanical origin of the analyzed samples, which can corroborate the high values in some compounds and relate them to the specific pollen of the plant.

Response: the sentence was deleted. Thank you very much for your suggestions in the near future we would like to do the relevant studies about the botanical origin, but due to extreme situation in the word we don t have the conditions to do this analysis. Moreover, we aimed to optimize the derivatization parameters and to investigate a simple and sensitive GC-FID method to determine fatty acids in the bee products.

Reviewer 3 Report

The manuscript entitled “Chemical Composition of Fatty Acids in Bee Products” described the comparison of the total fatty acids' concentration in four experimental bee products (honey, bee pollen, bee bread and propolis) collected from the Lithuania region. This manuscript should be accepted in Applied Science after revisions.

Abstract: Based on manuscript title, more information related to fatty acids in bee products should be included, instead optimization and validation. Line 23: “high recovery”, where is the information related to this parameter? In section 3.1., the selection of the best methylation parameters was performed based on the concentration of fatty acids.

All abbreviations should be described when used for the first time

Line 167: figure 2 should be Figure 2.

Line 204: figure 4 should be Figure 4

A Table containing the quantitative information should be added.

Reference 12, is incomplete.

Author Response

The authors would like to thank Reviewers for their comments and suggestions to improve the quality of the manuscript. Please find below our response to the remarks, the corresponding corrections of the manuscript have been made. Moreover, the stylistic corrections and additions as well as some new text were added on the basis of recommendations of reviewers.

Comments and suggestions to authors

Abstract: Based on manuscript title, more information related to fatty acids in bee products should be included, instead optimization and validation. Line 23: “high recovery”, where is the information related to this parameter? In section 3.1., the selection of the best methylation parameters was performed based on the concentration of fatty acids.

Response: the mistake has been corrected and instead of “high recovery” was changed to high amount.

All abbreviations should be described when used for the first time

Response: the abbreviations were described.

Line 167: figure 2 should be Figure 2.

Response: the mistake has been corrected.

Line 204: figure 4 should be Figure 4

Response: the mistake has been corrected.

A Table containing the quantitative information should be added.

Response: the quantitative information is shown in Figure 3.

Reference 12, is incomplete.

Response: the mistake has been corrected.

Round 2

Reviewer 2 Report

The text of the manuscript was improved and I consider it possible its publication in Applied Sciences. I only recommend a minor aspects:

Why do the authors use the term experimental (in bee produtcs) in all manuscript? Line 20, 24, 31....

Line 316-317 include it in the conclusions.

References should be checked, there are some journal errors.

Author Response

Response to the Editor and Reviewer’s comments on the manuscript

“Optimization and Validation of the GC/FID Method For the Quantification of Fatty Acids in Bee Products”

The authors would like to thank Reviewers for their comments and suggestions to improve the quality of the manuscript. Please find below our response to the remarks, the corresponding corrections of the manuscript have been made. Moreover, the stylistic corrections and additions as well as some new text were added on the basis of recommendations of reviewers.

Comments and suggestions to authors

Reviewer 2:

The text of the manuscript was improved and I consider it possible its publication in Applied Sciences. I only recommend a minor aspects:

Why do the authors use the term experimental (in bee produtcs) in all manuscript? Line 20, 24, 31....

Response: word ‘experimental’ was deleted.

Line 316-317 include it in the conclusions.

Response: the changes were done according to suggestions.

References should be checked, there are some journal errors.

Response: references have been corrected according to suggestions.
